# Investigation of Virulence-Related Markers in Atypical Strains of *Toxoplasma gondii* from Brazil

**DOI:** 10.3390/microorganisms13020301

**Published:** 2025-01-29

**Authors:** Júlia Gatti Ladeia Costa, Érica Santos Martins-Duarte, Lorena Velozo Pinto, Ramon Araujo de Castro Baraviera, Wagner Martins Fontes do Rego, Ricardo Wagner de Almeida Vitor

**Affiliations:** 1Departamento de Ciências Biológicas, Universidade do Estado de Minas Gerais, Ibirité 32412-190, Brazil; julia.costa@uemg.br; 2Departamento de Parasitologia, Universidade Federal de Minas Gerais, Belo Horizonte 31270-901, Brazil; emartinsduarte@gmail.com (É.S.M.-D.); nena.twist@hotmail.com (L.V.P.); ramonbaraviera@gmail.com (R.A.d.C.B.); wagner.fontesrego@gmail.com (W.M.F.d.R.)

**Keywords:** toxoplasmosis, rhoptry proteins, genotyping, cumulative mortality, PCR-RFLP

## Abstract

*Toxoplasma gondii* is an obligate intracellular protozoan parasite distributed worldwide that infects a wide range of warm-blooded animals, including humans. Recent studies sought to clarify the relationship between the alleles GRA15, ROP5, ROP16, ROP17, and ROP18 and the virulence of *T. gondii* isolates in mice. This work aims to analyze the variability of genes that express *T. gondii* virulence proteins of 103 strains. Most strains were virulent for mice (76/103–73.79%); within these, 30 were 100% lethal, and 46 caused a cumulative mortality range from 20% to 93%. For the GRA15 marker, most strains presenting allele 2 were non-lethal. For the ROP17 marker, allele 4 was associated with mortality, compared to allele 1. For the ROP18 marker, alleles 1 and 4 were associated with mortality, compared to alleles 2 and 3. A combined analysis of alleles showed low cumulative mortality when the strains presented alleles 3 and 1 for ROP18 and ROP16, respectively. On the other hand, allele 4 of ROP17 was a determinant for virulence when associated with ROP18 allele 3 and ROP16 allele 1. Our analysis shows that ROP18 is the primary determinant of the virulence of atypical strains in mice. Additionally, ROP17 genotyping should not be overlooked, as it has proven critical to enhance this prediction.

## 1. Introduction

Toxoplasmosis is a worldwide zoonosis that presents various clinical conditions, from asymptomatic infection to severe systemic manifestations. In immunocompetent individuals, the disease, in general, is asymptomatic. However, latent toxoplasmosis may be associated with neurological disorders such as schizophrenia [1]. In immunocompromised individuals, sequelae caused by *Toxoplasma gondii* have been observed in individuals with acquired immunodeficiency syndrome (AIDS), in treatment with immunosuppressive drugs, or in those who have received an organ or bone marrow transplant. In congenital toxoplasmosis, the parasite can cause abortion or the birth of children with ocular and neurological disorders [2,3].

Studies carried out with strains of *T. gondii* isolated from patients or animals from South America have shown a high rate of genetic polymorphism [4]. Recent studies sought to clarify the relationship between the alleles identified by polymerase chain reaction–restriction fragment length polymorphism (PCR-RFLP) of the protein virulence markers, such as dense granule protein 15 (GRA15), rhoptry protein 5 (ROP5), ROP16, ROP17, and ROP18, with the virulence of *T. gondii* isolates in mice [5]. Among these proteins, ROP16 and ROP17 were associated with the modulation of the host’s immune response by the parasite, and ROP18 and ROP5 were highlighted as the main determinants of *T. gondii* virulence in mice [5]. Rêgo et al. [6] demonstrated that in Brazilian atypical strains of *T. gondii*, the ROP18 protein, analyzed alone or in combination with ROP5, effectively determined the parasite’s virulence for mice. 

Different mouse strains, life cycle stages of the parasite, and routes of inoculation are used in *T. gondii* virulence assays. These limit comparisons considering genotype and virulence in mice and the integration of data [7]. Here, we analyze a large number of atypical isolates obtained in Brazil, including those isolated from humans, with a standardized methodology to evaluate the variability of genes that express *T. gondii* virulence proteins.

## 2. Materials and Methods

### 2.1. Toxoplasma gondii Strains

All 103 strains used in this work were previously isolated by our group (Appendix A) and have been maintained cryopreserved in N2 banks at the Laboratory of Toxoplasmosis at the Universidade Federal de Minas Gerais—UFMG, Brazil. According to Brazilian law, access to Brazilian genetic heritage was approved by SisGen protocols A3F9195, AD3C00F, AA14CC9, AB9C3CF, A375FCF, AC52DBB, and AD3745E.

### 2.2. PCR-RFLP Genotyping of T. gondii

PCR-RFLP protocols for GRA15, ROP16, and ROP18 were performed according to Dubey et al., 2014 [8]. PCR-RFLP protocols for ROP5 and ROP17 were performed according to Shwab et al. [5].

Genomic DNA samples from *T. gondii* strains were obtained according to Carneiro et al. [9] and provided by the Laboratory of Toxoplasmosis (UFMG). Briefly, DNA extraction of the tachyzoites was performed using the Promega Wizard genomic DNA purification kit following the manufacturer’s instructions. Genomic DNA stocks were stored at −20 °C to 4 °C before analysis. The RH88 (type I), ME49 (type II), PTG (II), VEG (type III), and MAS (atypical) strains were used as controls and references. The negative control was distilled water in the presence of primers. For each locus, PCR was performed on parasite genomic DNA using external primers, followed by a second round of PCR with nested primers using external primer products as templates. Primer sequences and restriction enzymes for PCR-RFLP genotyping of rhoptry gene loci ROP18, ROP5, ROP16, and ROP17 were the same as previously described [5,8]. The amplified products were digested using restriction endonucleases (New England BioLabs, Ipswich, MA, USA) specific for each marker according to the manufacturer’s instructions. The DNA of the digested products was purified by extraction with an equal volume of phenol/chloroform (1:1), subjected to polyacrylamide gel (5%) electrophoresis, stained with silver nitrate, and photographed.

### 2.3. Parasite Virulence Determination

Strain virulence information (Appendix A) was obtained from previously published data from bioassays in mice [6,9,10,11,12,13,14,15]. Five female BALB/c mice, six to eight weeks old, were inoculated intraperitoneally (i.p.) with 10, 100, or 1000 tachyzoites of each strain in 0.2 mL of PBS (pH 7.2). A total of 15 mice per strain were used. Five animals inoculated i.p. with PBS were maintained as negative controls. All efforts were made to minimize animal suffering during the study. Mouse mortality was observed for a period of 30 days. ELISA (anti-*T. gondii* IgG) was performed on all surviving mice to confirm infection. The survivor mice that did not seroconvert were excluded from the experiment. RH (virulent) and ME49 (nonvirulent) strains were used as references for comparison. According to Saraf et al. [7], cumulative mortality was calculated based on the number of mice that died divided by the number of mice infected, analyzing three sequential inoculation dosages, with the lowest dose resulting in only partial infection. The Animal Ethics Committee (CEUA) of the Universidade Federal de Minas Gerais (UFMG), Minas Gerais, Brazil, approved all experiments and procedures, including euthanasia and blood collection (CEUA Protocols: 013/2007, 128/2010, 266/2012, 067/2016, and 048/2018).

### 2.4. Statistical Analyses

The analysis of the correlation between allele types (GRA15, ROP5, ROP16, ROP17, and ROP18) and the median mouse cumulative mortality was performed using the non-parametric Mann–Whitney U test (individual comparisons) and the Kruskal–Wallis test (group analysis) followed by Dunn’s test for multiple comparisons. Statistical correlation between lethality and genotype was analyzed using Pearson chi^2^. Analyses were performed using GraphPad Prism version 5.0 (GraphPad Software, La Jolla, CA, USA), and *p* < 0.05 was considered significant.

## 3. Results

In total, we studied 103 atypical strains isolated from four different Brazilian states: Espírito Santo state (ES) and Minas Gerais state (MG), both located in Southeast Brazil, with a predominantly tropical climate, and Piauí state (PI) and Rio Grande do Norte state (RN), both located in the Northeast region of Brazil with a predominantly semi-arid climate (Figure 1A). Considering the hosts, 29 strains were obtained from humans (28.16%), 29 (28.16%) from chickens, 21 (20.39%) from pigs, 10 (9.71%) from goats, 8 (7.77%) from dogs, and 6 (5.83%) from wild birds (Appendix A). Most strains showed some virulence level for mice (76/103–73.79%); within these, 30 were 100% lethal, and 46 caused a cumulative mortality ranging from 20% to 93%. Within the 103 strains, previous studies showed that 39 different genotypes were identified, with 26 strains of unique genotypes, 17 identified as ToxoDB #163, 14 as #11(BrII), 7 as #108, 6 as #146, 6 as #206, 5 as #8 (BrIII), 5 as #109, 4 as #13, 4 as #24, 3 as #57, 2 as #6 (BrI), 2 as #41, and 2 as #19 (Appendix A).

The atypical strains presented diverse combinations of virulence factors. Representative gel images of PCR-RFLP genotyping are presented in Appendix A. There was no difference between the accumulated mortality from strains obtained in different geographic regions (*p* = 0.2083). The diversity of alleles found for the studied markers GRA15, ROP5, ROP16, ROP17, and ROP18 in strains obtained from different geographic regions is shown in Figure 1B. To validate the correlation of the allele types for GRA15, ROP5, ROP16, ROP17, ROP18, and combinations of ROP18/ROP5 with the virulence of *T. gondii* strains in mice, we plotted allele types against the cumulative mortality (Figure 2). The cumulative mortality was calculated from published data (Appendix A). Despite a large variation in the percentage of cumulative mortality within allele types, the differences between the medians were significant for some virulence factor types. For the GRA15 marker, most strains presenting allele 2 showed cumulative mortality of 0%, and the median cumulative mortality in isolates that presented alleles 1 or 3 was around 90% (Figure 2A). For the ROP17 marker, we observed that the median cumulative mortality of the isolates presenting allele 4 was significantly higher than that of allele 1 (Figure 2B). For the ROP16 and ROP5 markers, there were no differences in the cumulative mortality within the different alleles (Figure 2C,E).

The isolates that presented allele 1 of the ROP18 marker had 100% cumulative mortality, a significantly different value from those that had the 2 or 3 (*p* < 0.001) allele (Figure 2D). The median cumulative mortality for the isolates with alleles 1 or 4 of the ROP18 marker was higher than that of isolates with alleles 2 or 3. Equally, analysis of the combined alleles of ROP18/ROP5 with mouse virulence indicated that the combination had an overall stronger correlation. Six different combinations were identified (Figure 2F). Some combination segregation with high or low mouse virulence was statistically significant.

The relation of alleles 4 and 3 of ROP18 with high or low cumulative mortality, respectively, was evident when a combined analysis of alleles of ROP18 and ROP16 was performed. While no significant statistical difference was seen between the combinations of ROP18 allele 4 with ROP16 alleles 1 or 2 (Figure 3B), a significant difference was seen when the strains presented alleles 3 and 1 for ROP18 and ROP16, respectively, compared to those with ROP18 allele 4 (Figure 3B). The relation of allele 4 of ROP18 with cumulative mortality in strains presenting ROP16 allele 2 was irrespective of the ROP17 allele, as no significant difference was seen between the presence of ROP17 alleles 4 and 3 (Figure 3E). However, allele 4 of ROP17 was a determinant for the virulence of strains with ROP18 allele 3 and ROP16 allele 1 (Figure 3D). In the strains harboring alleles 3 and 1 of ROP18 and ROP17, respectively, the presence of alleles 1/3 or 2 of GRA15 did not influence virulence (Figure 3C).

To identify alleles that could be possible molecular markers of lethal strains, we also analyzed the frequency of alleles in two groups of strains according to their lethality (Table 1). Only 2 isolates from 50 that presented ROP18 allele 4 were non-lethal. ROP18 allele 1 and allele 2 were only found in lethal and non-lethal strains, respectively. Allele 4 of ROP5 and allele 3 of ROP17 were identified only in lethal strains, but all of them also harbored ROP18 allele 4. Strains harboring ROP17 allele 4 also showed a higher frequency of lethal phenotype. A higher frequency of non-lethal phenotype was observed in strains presenting GRA15 allele 2, ROP18 allele 3, and ROP5 allele 1 (Table 1). However, all strains with ROP5 allele 1 also harbored ROP18 allele 3, and the non-lethal phenotype is possibly associated with ROP18 alleles. Equally, most strains with ROP16 allele 2 presented a lethal phenotype; however, those also harbored ROP18 allele 4. In conclusion, our analysis shows that ROP18 is the major determinant of the virulence of atypical strains in mice.

## 4. Discussion

*Toxoplasma gondii* is a successful protozoa parasite due to its ability to manipulate host cell behavior [16]. Rhoptry proteins (ROPs) function at the parasitophorous vacuole to inhibit the recruitment of immunity-related GTPases (IRGs) and guanylate-binding proteins (GBPs), playing a critical role in host–parasite interactions [17,18]. ROP5 and ROP18 work synergistically to block innate immune mechanisms triggered by IFN-γ in murine hosts and serve as predictors of strain virulence in mice [5,19]. Consistent with our findings, previous studies have shown that the combined analysis of ROP18 and ROP5 can predict high virulence in mice from Brazilian strains isolated from pigs, goats, and even cases of human congenital toxoplasmosis [6,20]. However, most strains analyzed in this study exhibited ROP5 allele 3, which made it difficult to assess the contribution of this allele to the observed virulence.

In this work, we found ROP18 allele 4 in 49 isolates; in contrast, only 9 isolates had allele 1. Additionally, the combination of ROP18 allele 2/ROP5 allele 3 had not been previously reported, and strains carrying these alleles showed low cumulative mortality. The ROP18 allele 4/ROP5 allele 4 combination was first identified in *T. gondii* samples from Misiones province, Argentina, leading the authors to propose a possible regionalization of the ROP18/ROP5 profile [21]. However, in our study, we analyzed many strains and detected the same ROP18 allele 4/ROP5 allele 4 combination in four strains from Minas Gerais, Brazil. This finding underscores the importance of analyzing isolates from different regions to validate the predictive value of the ROP18/ROP5 profile for *T. gondii* virulence in mice, especially considering that standardized bioassays in mice are time-consuming, costly, and not feasible in all laboratories [7,22].

ROP5 forms complexes with the ROP18 and ROP17 kinases, working together to regulate acute virulence in mice [23]. ROP16 allele 1/2 can phosphorylate STAT3 and STAT6, leading to the alternative activation of macrophages (M2), which promotes parasite replication within host cells [19]. When analyzed together, our results indicate that ROP18 alleles 1 and 4, irrespective of the ROP5 or ROP16 background, are strongly associated with the virulence of Brazilian strains, showing a high predictive value for cumulative mortality in mice.

In contrast, ROP18 allele 3 is associated with lower virulence and may be influenced by the ROP17 background. Etheridge et al. [23] showed that while ROP17 is not directly responsible for mortality in mice, it contributes to the enhanced virulence associated with the ROP18/ROP5 combination. Hamilton et al. [24] suggested that ROP17 may play a role in determining virulence, which could explain cases where strains predicted to be non-lethal based on the ROP18/ROP5 allele combination are found to be virulent in mouse bioassays. This makes ROP17 a priority gene for genotyping analysis [25]. Thus, determining virulence may be more complex when examining only a few alleles.

GRA15 allele 2 emerged as an important marker in the non-lethal strains identified in this study. In clonal type II strains, it facilitates NF-kB nuclear translocation, induces macrophages toward a classically activated (M1) phenotype, and enhances the host’s innate immunity against *T. gondii* infection [26]. Notably, the only lethal strain with GRA15 allele 2 also harbored ROP18 allele 4 and ROP17 allele 4, both key virulence markers, despite the presence of ROP16 allele 2.

Like the current study, most research has focused on the mechanisms by which ROP proteins confer virulence in mice. However, how those virulence factors would impact the outcome of *T. gondii* infection in humans still needs to be discovered [27]. However, type I and atypical strains are frequently associated with symptomatic and severe forms of toxoplasmosis [28,29,30,31,32], and some studies shown the possible association of ROP18 with congenital and ocular diseases in humans [33,34,35].

To date, strain virulence cannot be reliably assessed using PCR-RFLP genotyping alone, except for the three archetypal strains [36]. Virulence protein polymorphisms were accessed in our study using the PCR-RFLP technique. This methodology limits the detection of single nuclear polymorphisms (SNPs) that are recognized by restriction enzymes and may not be sufficient to detect all polymorphisms in key genes that contribute to phenotypic and pathogenic differences in *T. gondii* strains [16]. So, a future perspective is to sequence the virulence proteins of isolates that presented alleles related to virulence but were non-lethal to mice or vice versa. However, a growing body of research, including our study, indicates that the combination of ROP18 and ROP5 loci offers a strong predictive capacity for virulence in mice, as proposed by Shwab et al. [5].

Our analysis shows that ROP18 is the major determinant of the virulence of Brazilian atypical strains in mice, and highlights the higher prevalence of strains with allele 4. However, how that allele would modulate the immune system in mice or humans has yet to be found. Additionally, ROP17 genotyping should not be overlooked, as it has proven to be a key factor in enhancing virulence prediction. Establishing the fundamental roles of these genetic markers in identifying the virulence of *T. gondii* will allow, in the future, the identification of virulent strains circulating in the environment, especially in livestock. Consequently, more appropriate measures to control the parasite will be adopted.

A limitation of our study is that we evaluated 103 isolates of *T. gondii* of only 39 genotypes. Thus, it is necessary to study more strains, with different genotypes obtained from both animals and humans, from other Brazilian regions and worldwide, to confirm our findings. We expect this work to stimulate other research groups to include similar or other analyses with more strains of identical or different genotypes from different world regions. Thus, over time, this analysis will gain strength and will support whether ROP18 is the major virulence determinant of *T. gondii* in mice and, potentially, in humans.

## Figures and Tables

**Figure 1 microorganisms-13-00301-f001:**
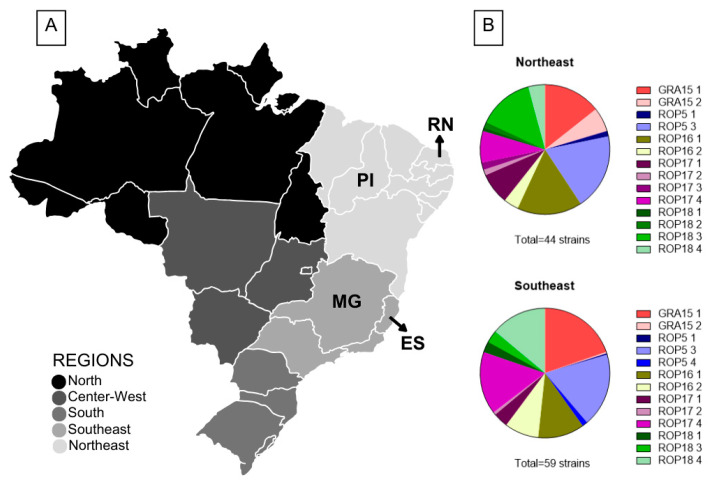
Distribution of *Toxoplasma gondii* strains according to the Brazilian regions from which they were isolated. (**A**) Map of Brazil showing the four states where the 103 *Toxoplasma gondii* strains were isolated: ES (Espírito Santo state), MG (Minas Gerais state), PI (Piauí state), and RN (Rio Grande do Norte state). In total, 5 strains were obtained from chickens from ES; 54 strains were obtained from MG (29 from humans—25 from peripheral blood of newborns with congenital toxoplasmosis and 4 from amniotic fluid; 8 from dogs; 11 from chickens; and 6 from free-living wild birds); 25 strains were obtained from PI (16 from pigs and 9 from goats); and 19 strains were obtained from RN (13 from chickens, 5 from pigs, and 1 from goat). (**B**) Distribution of alleles of the virulence protein genes GRA15, ROP5, ROP16, ROP17, and ROP18 identified by PCR-RFLP in 103 strains isolated in the Northeast and Southeast regions of Brazil.

**Figure 2 microorganisms-13-00301-f002:**
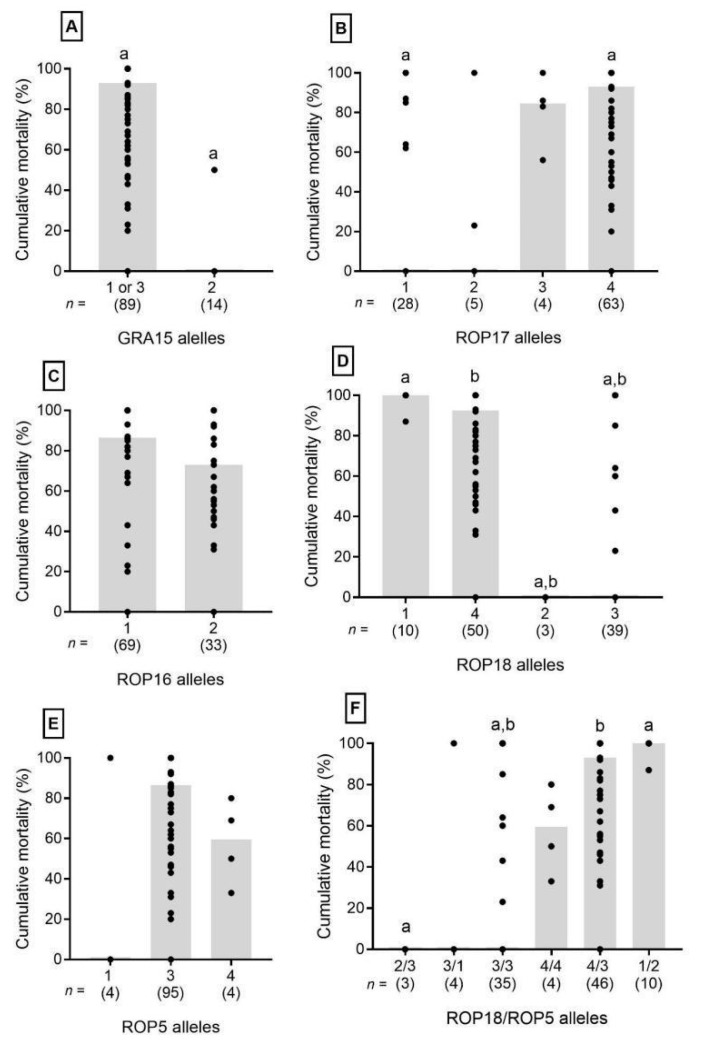
Virulence of *T. gondii* strains in mice, categorized by dense granule GRA15 gene and rhoptry genes ROP5, ROP16, ROP17, and ROP18 allele types. Cumulative mouse mortality for 103 *Toxoplasma gondii* strains with published bioassay results was categorized for alleles of GRA15 (**A**), ROP17 (**B**), ROP16 (**C**), ROP18 (**D**), ROP5 (**E**), and the ROP18/ROP5 (**F**) combinations. The percentage of mouse cumulative mortality was plotted per allele type, with each dot representing the percent mortality reported for an individual parasite strain. The gray bars indicate the median cumulative mortality for each allele type. The number (n) of strains representing each allele type is in parentheses below allele identification. Different letters above allele data (a or b) indicate statistical differences as determined by Mann–Whitney U tests. Differences between data are significant at *p* < 0.001 for GRA15 and ROP18, *p* = 0.003 for ROP17, and *p* < 0.05 for the ROP18/ROP5 allele combinations.

**Figure 3 microorganisms-13-00301-f003:**
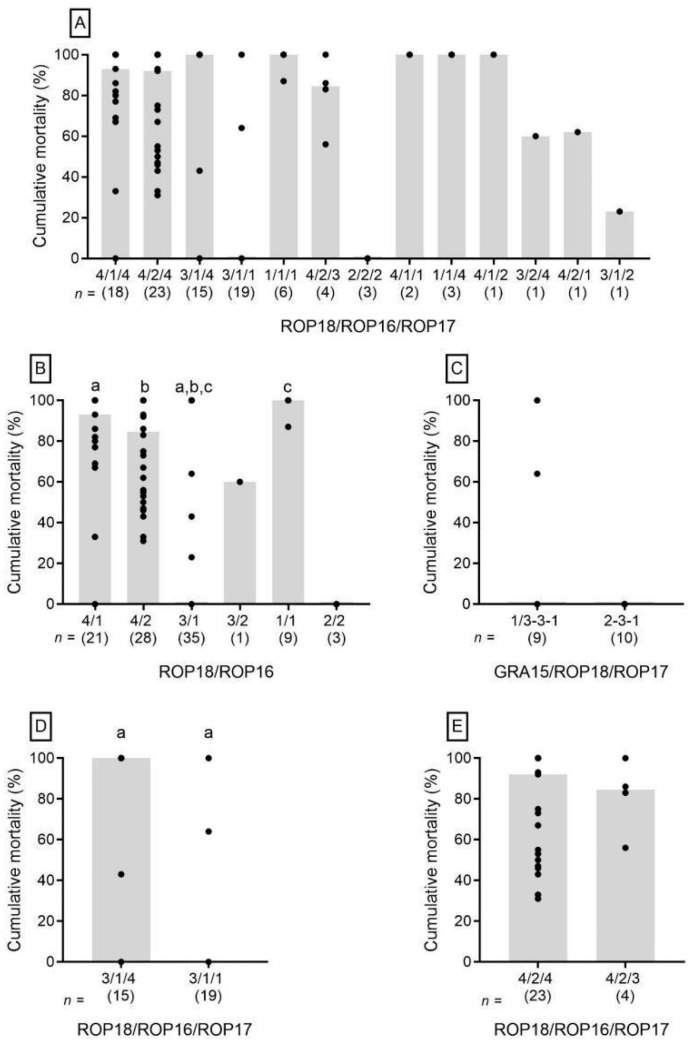
Cumulative mouse mortality for 103 *Toxoplasma gondii* strains categorized for ROP18/ROP17/ROP16 (**A**,**D**,**E**) ROP18/ROP16 (**B**), and GRA15/ROP18/ROP17 (**C**) allele combinations. Percentage of mouse cumulative mortality is plotted against allele type, with each individual dot representing the percentage of mortality reported for an individual parasite strain. The gray bars indicate the median cumulative mortality for each allele type. The number (n) of strains representing each allele type is given in parentheses below allele identification. Different letters above allele data (a, b, or c) indicate statistical differences as determined by Mann–Whitney U tests. Differences between data are significant at *p* < 0.0001 for the ROP18/ROP16 allele combinations (**B**) and at *p* = 0.0004 for the ROP18/ROP16/ROP17 allele combinations (**D**).

**Table 1 microorganisms-13-00301-t001:** Distribution of alleles of the virulence protein genes GRA15, ROP5, ROP16, ROP17, and ROP18 identified by PCR-RFLP in Brazilian atypical strains according to lethality in mice.

	Lethal Strains	Non-Lethal Strains		
GRA15	n (*f*)	n (*f*)	Total	*p* value *
1 or 3	75 (98.7%)	14 (51.9%)	89 (86.4%)	<0.001
2	1 (1.3%)	13 (48.1%)	14 (13.6%)	
Total	76	27	103	
ROP5				
1	1 (1.3%)	3 (11.1%)	4 (3.9%)	0.04
3	71 (93.4%)	24 (88.9%)	95 (92.2%)	
4	4 (5.3%)	0 (0.0%)	4 (3.9%)	
Total	76	27	103	
ROP16				
1	46 (60.5%)	23 (88.5%)	69 (67.6%)	0.007
2	30 (39.5%)	3 (11.5%)	33 (32.4%)	
Total	76	26	102 ^a^	
ROP17				
1	11 (15.1%)	17 (63.0%)	28 (28.0%)	<0.001
2	2 (2.7%)	3 (11.1%)	5 (5.0%)	
3	4 (5.5%)	0 (0.0%)	4 (4.0%)	
4	56 (76.7%)	7 (25.9%)	63 (63.0%)	
Total	73	27	100 ^b^	
ROP18				
1	10 (13.3%)	0 (0.0%)	10 (9.8%)	<0.001
2	0 (0.0%)	3 (11.1%)	3 (2.9%)	
3	17 (22.7%)	22 (81.5%)	39 (38.2%)	
4	48 (64.0%)	2 (7.4%)	50 (49.0%)	
Total	75	27	102 ^a^	

^a^ incomplete ROP16 and ROP18 genotyping for one strain. ^b^ incomplete ROP17 genotyping for three strains. * Pearson’s chi^2^.

## Data Availability

The raw data supporting the conclusions of this article will be made available by the authors on request.

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
