# Peer review of "Investigation of Virulence-Related Markers in Atypical Strains of *Toxoplasma gondii* from Brazil"

_microorganisms, 2025, doi:10.3390/microorganisms13020301_

Round 1
Reviewer 1 Report
Comments and Suggestions for Authors
In this manuscript, the authors have conducted an analysis of 103 Toxoplasma gondii isolates from various animal and human sources to assess the variability of genes expressing T. gondii virulence proteins and their correlation with virulence in mice. The findings of this study are expected to enhance more understanding of T. gondii strain variability and its implications for disease severity. However, some suggestions need to be addressed to improve the manuscript.
1. The manuscript contains grammatical errors and typographical mistakes that require thoroughly correction during the revision process. For example, "T. gondii," "GRA," "ROP," and others should be given their full names upon their initial mention. “T. gondii”, “GRA”, “ROP” and other abbreviations need to be full name at first use. Lines 72-75, there are spelling errors with percentages that need to be corrected.
2. The Abstract lacks a background context on Toxoplasma gondii. The authors should provide a brief overview of the parasite and its significance in the field of study to better orient the reader.
3. The authors should re-evaluate the statistical methods used, ensuring that they account for multiple comparisons. If necessary, appropriate corrections should be applied to address this concern.
4. The authors should describe the limitations of the current study, which should be discussed in the Discussion section.
Author Response
In this manuscript, the authors have conducted an analysis of 103 Toxoplasma gondii isolates from various animal and human sources to assess the variability of genes expressing T. gondii virulence proteins and their correlation with virulence in mice. The findings of this study are expected to enhance more understanding of T. gondii strain variability and its implications for disease severity. However, some suggestions need to be addressed to improve the manuscript.
- The manuscript contains grammatical errors and typographical mistakes that require thoroughly correction during the revision process. For example, "T. gondii," "GRA," "ROP," and others should be given their full names upon their initial mention. “T. gondii”, “GRA”, “ROP” and other abbreviations need to be full name at first use. Lines 72-75, there are spelling errors with percentages that need to be corrected.
1-Response: We are thankful to the Reviewer 1 for comments. Manuscript revisions were highlighted in red. During the review we identified and corrected some grammatical errors. Errors with percentages were corrected. The full names of Toxoplasma gondii, PCR-RFLP, GRA and ROP have been added.
- The Abstract lacks a background context on Toxoplasma gondii. The authors should provide a brief overview of the parasite and its significance in the field of study to better orient the reader.
2-Response: The Abstract has been improved. The 200-word limit for the abstract makes it difficult to include more information about T. gondii. The background context on Toxoplasma gondii is described in the introduction.
- The authors should re-evaluate the statistical methods used, ensuring that they account for multiple comparisons. If necessary, appropriate corrections should be applied to address this concern.
3-Response: We consider the statistical methods applied in the study to be adequate for a coherent and effective analysis. For better understanding we added the post test information for multiple comparisons.
- The authors should describe the limitations of the current study, which should be discussed in the Discussion section.
4-Response:The following sentences were added to the discussion, concerning limitations of the current study :
- The virulence protein polymorphisms were accessed in our study using the PCR-RFLP technique. This methodology limits the detection of single nuclear polymorphisms (SNPs) that are recognized by restriction enzymes and may not be sufficient to detect all polymorphisms in key genes that contribute to phenotypic and pathogenic differences of T. gondii strains [16]. So, a future perspective is to sequence the virulence proteins of isolates that presented alleles related to virulence but were non-lethal to mice or vice versa.
-A limitation of our study is that we evaluated 103 isolates of T. gondii of only 39 genotypes. Thus, it is necessary to study more strains, with different genotypes obtained from both animals and humans, from other Brazilian regions and worldwide, to confirm our findings. We expect this work to stimulate other research groups to include similar or other analyses with more strains of identical or different genotypes from different world regions. Thus, over time, this analysis will gain strength and will support if ROP18 is the major virulence determinant of T. gondii in mice and, who knows, in humans.
Reviewer 2 Report
Comments and Suggestions for Authors
The objective of this study is to investigate 103 isolates from both animals and humans, focusing on the genetic diversity of genes that encode virulence factors of T. gondii, and to explore their potential relationship with virulence in mice.
Major Comments:
1. The Materials and Methods section of the article needs to be improved. Although authors cite published articles in the material methods section, they should also be appropriately described. And the materials and methods should continue to be subdivided, and each part of the materials and methods should have corresponding secondary headings to make it easier for readers to find and read.
2. Line 61 “previously published data of bioassay in mice”. Authors should count or describe key mouse information: breed of mouse (C57BL/6 or BALB/c), weekly age, mode of infection with Toxoplasma gondii.
3. The date of death of mice infected with Toxoplasma gondii is also of great reference value. Did the authors analyze this part of the data?
minor comments:
1.The full name of PCR-RFLP should be indicated when this word appears for the first time.
Author Response
The objective of this study is to investigate 103 isolates from both animals and humans, focusing on the genetic diversity of genes that encode virulence factors of T. gondii, and to explore their potential relationship with virulence in mice.
Major Comments:
- The Materials and Methods section of the article needs to be improved. Although authors cite published articles in the material methods section, they should also be appropriately described. And the materials and methods should continue to be subdivided, and each part of the materials and methods should have corresponding secondary headings to make it easier for readers to find and read.
1.Response: We are thankful to the Reviewer 2 for comments. Manuscript revisions were highlighted in red.The Materials and Methods section have been improved. We do not consider necessary to cite further details of RFLP, in addition to those included in the new version, since this methodology is well described in previous works, as included in references: Dubey, et al. 2014. doi:10.1016/j.vetpar.2013.11.001; Shwab et al. 2016. doi:10.1016/j.ijpara.2015.10.005.
- Line 61 “previously published data of bioassay in mice”. Authors should count or describe key mouse information: breed of mouse (C57BL/6 or BALB/c), weekly age, mode of infection with Toxoplasma gondii.
2.Response: Experiments with mice have been described.
- The date of death of mice infected with Toxoplasma gondii is also of great reference value. Did the authors analyze this part of the data?
3.Response: The date of death of infected mice was not analyzed in this study.
minor comments:
- The full name of PCR-RFLP should be indicated when this word appears for the first time.
1.Response: The full name of PCR-RFLP has been added.
Reviewer 3 Report
Comments and Suggestions for Authors
The manuscript by Júlia Gatti Ladeia Costa et al. titled "Virulence-related markers investigation in atypical strains of Toxoplasma gondii from Brazil" is a brief communication devoted to allelic RFLP-PCR typing of 103 T. gondii isolates across 5 loci and the association of these alleles with virulence and mortality in mice. The manuscript is quite interesting and well-written.
There are just a few minor notes:
- In the figure legends for Figures 1 and 2, an explanation should be added stating that in Fig. 1A, 1B, 1C, etc., and also 2A, 2B, 2C...
- Toxoplasma gondii should be spelled out in full at the first mention (line 32), not in the middle of the text (line 163).
- Line 345: This is too convoluted. The journal name should simply be "Infect. Genet. Evol."
Author Response
The manuscript by Júlia Gatti Ladeia Costa et al. titled "Virulence-related markers investigation in atypical strains of Toxoplasma gondii from Brazil" is a brief communication devoted to allelic RFLP-PCR typing of 103 T. gondii isolates across 5 loci and the association of these alleles with virulence and mortality in mice. The manuscript is quite interesting and well-written.
There are just a few minor notes:
1- In the figure legends for Figures 1 and 2, an explanation should be added stating that in Fig. 1A, 1B, 1C, etc., and also 2A, 2B, 2C...
1-Response: We are thankful to the Reviewer 3 for comments. Manuscript revisions were highlighted in red. The legends of Figures 1 and 2 have been improved
2- Toxoplasma gondii should be spelled out in full at the first mention (line 32), not in the middle of the text (line 163).
2-Response: The full name Toxoplasma gondii has been added at the first mention.
3- Line 345: This is too convoluted. The journal name should simply be "Infect. Genet. Evol."
3-Response: The journal name has been corrected.
Reviewer 4 Report
Comments and Suggestions for Authors
This study provides a valuable contribution to understanding the genetic markers associated with virulence in atypical Toxoplasma gondii strains from Brazil. By analyzing 103 isolates from animals and humans, it examines the variability of key virulence-related genes (GRA15, ROP5, ROP16, ROP17, ROP18) and their associations with mouse mortality. The findings highlight the critical role of ROP18, as well as the importance of combined genetic analyses for accurate virulence prediction. The study is well-structured but could benefit from additional details and clarifications in several areas, as outlined below.
Suggestions:
Include a geographical map showing the locations where the 103 isolates were collected. This will provide context for the diversity and potential regional clustering of virulence traits among the isolates. Such a map could offer insights into environmental or ecological factors influencing genetic variability and virulence.
Clearly state the criteria used to include isolates in the study. Specify whether factors such as host species (animal or human), clinical presentation, or geographical origin influenced inclusion.
Provide a detailed statement on the ethical considerations for the use of mice in this study. Include the total number of mice used, the distribution across the 103 isolates, and any humane endpoints implemented.
Expand the description of the genetic analysis methods for determining allele types for GRA15, ROP5, ROP16, ROP17, and ROP18. Include the following: DNA extraction techniques, specific primers or sequencing platforms used, any quality control measures and analytical approaches for associating alleles with virulence.
Discussion of implications:
How these findings could influence public health strategies or veterinary interventions in Brazil.
Potential applications of these genetic markers in predicting strain virulence.
I commend the authors on their work and look forward to seeing the revised manuscript.
Author Response
This study provides a valuable contribution to understanding the genetic markers associated with virulence in atypical Toxoplasma gondii strains from Brazil. By analyzing 103 isolates from animals and humans, it examines the variability of key virulence-related genes (GRA15, ROP5, ROP16, ROP17, ROP18) and their associations with mouse mortality. The findings highlight the critical role of ROP18, as well as the importance of combined genetic analyses for accurate virulence prediction. The study is well-structured but could benefit from additional details and clarifications in several areas, as outlined below.
Suggestions:
1 Include a geographical map showing the locations where the 103 isolates were collected. This will provide context for the diversity and potential regional clustering of virulence traits among the isolates. Such a map could offer insights into environmental or ecological factors influencing genetic variability and virulence.
1 Response: We are thankful to the Reviewer 4 for comments. Manuscript revisions were highlighted in red. A geographical map (new Figure 1) was included as suggested.
2 Clearly state the criteria used to include isolates in the study. Specify whether factors such as host species (animal or human), clinical presentation, or geographical origin influenced inclusion.
2 Response: There was no criteria for inclusion of isolates. All isolates obtained by our research group and available in our cryobank (Laboratory of Toxoplasmosis - Federal University of Minas Gerais – UFMG) were used, regardless of host species, clinical presentation, or geographical origin.
3 Provide a detailed statement on the ethical considerations for the use of mice in this study. Include the total number of mice used, the distribution across the 103 isolates, and any humane endpoints implemented.
3 Response: Experiments with mice have been described, as well as the ethical considerations for the use of mice.
4 Expand the description of the genetic analysis methods for determining allele types for GRA15, ROP5, ROP16, ROP17, and ROP18. Include the following: DNA extraction techniques, specific primers or sequencing platforms used, any quality control measures and analytical approaches for associating alleles with virulence.
4 Response: The Materials and Methods section have been improved. We do not consider necessary to cite further details of RFLP in addition to those included in the new version, since this methodology is well described in previous works, as included in references.
Discussion of implications:
5 How these findings could influence public health strategies or veterinary interventions in Brazil.
5 Response: This issue was addressed in the discussion with the inclusion of the following sentence:
Establishing the fundamental roles of these genetic markers in identifying the virulence of T. gondii will allow, in the future, the identification of virulent strains circulating in the environment, especially in livestock. Consequently, more appropriate measures to control the parasite will be adopted.
6 Potential applications of these genetic markers in predicting strain virulence.
6 Response: The scarce studies evaluating cases of human symptomatic toxoplasmosis and genetic markers of the strains involved still make it difficult to apply these markers to predict the virulence of the strain. This issue was addressed at the of the discussion.
I commend the authors on their work and look forward to seeing the revised manuscript.
Round 2
Reviewer 4 Report
Comments and Suggestions for Authors
Congratulations!